# Self-referential encoding of source information in recollection memory

Ross Lawrence[1]*, Xiaoqian J. Chai[2]

1 Department of Neurology, Laboratory X, Johns Hopkins University, Baltimore, MD, United States of America, 2 Department of Neurology and Neurosurgery, Laboratory X, McGill University, Montreal, QC, Canada

* rlawre18@jhmi.edu

## Abstract

Information that is encoded in relation to the self has been shown to be better remembered, yet reports have disagreed on whether the memory benefit from self-referential encoding extends to source memory (the context in which information was learned). In this study, we investigated the self-referential effect on source memory in recollection and familiarity-based memory. Using a Remember/Know paradigm, we compared source memory accuracy under self-referential encoding and semantic encoding. Two types of source information were included, a "peripheral" source which was not inherent to the encoding activity, and a source information about the encoding context. We observed the facilitation in item memory from self-referential encoding compared to semantic encoding in recollection but not in familiarity-based memory. The self-referential benefit to source accuracy was observed in recollection memory, with source memory for the encoding context being stronger in the self-referential condition. No significant self-referential effect was observed with regards to peripheral source information (information not required for the participant to focus on), suggesting not all source information benefit from self-referential encoding. Self-referential encoding also resulted in a higher ratio of "Remember/Know" responses rate than semantically encoded items, denoting stronger recollection. These results suggest self-referential encoding creates a richer, more detailed memory trace which can be recollected later on.

## 1 Introduction

Self-referential encoding, when information is encoded with reference to the self (e.g. "What is your opinion of this object?", "Does this adjective describe you?"), has been shown to lead to better memory performance compared to other encoding strategies, including semantic and other-referent encoding [1, 2]. This facilitation in memory from self-referential encoding is known as the Self-Referencing Effect (SRE). The improvement in memory performance due to SRE is not limited to particular types of stimuli. A meta-analysis of SRE research reported that although approximately 80% of all studies used personality trait words, SRE has been documented across a variety of stimuli, from trait adjectives and nouns [2], to photographic objects

**Data Availability Statement:** All behavioral result files are available from the corresponding OSF database (https://osf.io/w43my/).

**Funding:** X.C.: Therapeutic Cognitive Neuroscience Fund Grant Number: 80026224. The funders had no role in study design, data collection and

analysis, decision to publish, or preparation of the manuscript.

**Competing interests:** The authors have declared that no competing interests exist.

[3]. Proposed theoretical explanations for the SRE posit that there exist well-established networks of knowledge/memories related to the self that self-referential processing taps into, allowing for more organized and elaborate processing than other information processing methods [2, 4–6].

Historically, research into SRE has focused more on item recognition, with fewer studies investigating the accompanying source information of the item being encoded. Source information pertains to any and all features that, collectively, describe the conditions under which the memory was formed. This information can include spatial, temporal, visual, and/or the delivery method of the stimuli [7]. Several studies have examined self-referential encoding in source memory paradigms but the results have been inconsistent. Beneficial SRE on item and source memory was observed as improved accuracy in determining the background image displayed with the object and/or the proper encoding prompt [8–12]. A recent study reported self-referential facilitation of source information involving the spatial location of words, but not the color the words were displayed as [13]. Another study by Durbin, Mitchell, and Johnson [14] suggested that the SRE on source memory may depend on the valence (positive/neutral/negative association) of the items being processed. While self-referential encoding enhanced item recognition for positive, negative and neutral words, source memory (remembering what prompt accompanied the word, "Me?" or "Story?") was facilitated by self-referential encoding only in positive words, not in neutral or negative words. When the experiment was repeated with pictures, self-referential encoding actually resulted in worse source memory for neutral and negative pictures compared to non-self-referentially encoded pictures. While it is known that both positive and negative stimuli are better remembered than neutral stimuli [15–17], the interaction between valence and SRE has not been consistently reported. In D'Argembeau, Comblain, et al. [18] the SRE only improved the retrieval of positive emotional information, not the negative information, and only influencing free recall but not recognition. Fossati, Graham, et al. [19] observed a contradictory phenomenon, with young adults recognizing more negative words than positive ones regardless of the encoding condition. Other studies have found no significant interaction between valence and SRE [17, 20].

One key distinction to be made with reference to source memory is whether the source information being tested for is inherent to the encoding activity as opposed to additional encoding context. The majority of studies regarding source memory use tasks in which the source information is required to be processed by the participant, for example, what encoding activity (e.g., self-encoding versus semantic encoding) was coupled with the stimuli [3, 14, 21]. The few studies that did monitor additional source information still required participants to explicitly allocate attention to the source information. For example, Leshikar and Duarte [9] conducted a study in which the participants were show images on one of two backgrounds and asked "Is this object-scene pairing pleasant?" (self-referential) or "Is the dominant color of the object found in the background?" (self-external). The resulting source memory performance was based off of the recollection of the prompt and the background, two pieces of information that the participant had to pay attention to in order to perform the task. SRE's influence on peripheral information, information not necessary to properly preform the task (such as background not referenced by any prompt, or color of another object presented with the stimuli), during encoding has sparsely been tested, if at all.

Another line of research has focused on SRE in recollection vs. familiarity-based memory, which measures an individual's subjective recollection. Subjective recollection refers to when a person determines whether or not they are able to remember any episodic details while recalling information. The Remember/Know paradigm, where "Remember" denotes a conscious recollection of specific details relating to the item and accompanying details of its prior occurrence, and "Know" denotes only a familiarity without said episodic information, has

commonly been used to investigate recollection [22, 23]. Conway and Dewhurst [24] reported that adults had similar overall recognition rates for both self-referentially and semantically encoded words. However, analysis of recognition in terms of numbers of correct "Remember"/"Know" responses revealed significantly higher ratio of "Remember" responses to "Know" responses for self-referential encoding compared to non-self-referential encoding. Similar results were found by later studies [25, 26]. This supports a possible interaction between SRE and subjective recollection independent of overall item recognition. The link between SRE and subjective recollection may be invariant to the stimuli valence as well, as Lalanne, et al. [21] found that SRE improved recognition performance in young adults and significantly influenced the proportion of "Remember" responses, with the proportion not varying according to whether the adjectives were positive or negative.

It is so far unclear how the subjective recollection experience influences SRE on source memory. Recollection, relative to familiarity-based memory, presumably contains more source details. The goal of the current study was to investigate the SRE on source memory in recollection, with self-referential encoding being compared with semantic encoding in an incidental encoding source memory task, using the "Remember/Know" paradigm [22]. Our design included two types of source information, a "peripheral" source which was not inherent to the encoding activity, and source information about the encoding context (encoding question). This would allow us to examine whether self-referential encoding has different effects on these different types of source information.

## 2 Methods

### 2.1 Ethics statement

This study was approved by the Johns Hopkins School of Medicine IRB, Approval Number: IRB00151734. IRB approved written consent was collected from each participant. Individuals aged 18–35 were recruited for participation in this study due to its focus on memory in neurotypical adults.

### 2.2 Participants

52 healthy adults between 18 and 35 years of age participated in the study (25 females and 27 males, mean age = 23.64 ± 4.94). Participants were recruited from the Johns Hopkins University community and the Baltimore area. All participants were native English speakers, right-handed, had normal or corrected-to-normal vision, with no history of psychiatric, neurological, or developmental disorder. Informed consent from was obtained prior to the study. All participants were given a Kaufman Brief Intelligence Test-2 (K-bit) during the visit. The range of participant IQ scores were 98 to 138 (mean = 119.33 ± 9.92).

### 2.3 Behavioral task

**2.3.1 Encoding task.** The encoding part of the study took place as part of a larger study involving the collection of MRI data using an MRI scanner. Before going inside the MRI scanner, all participants signed the consent forms and were thoroughly explained the encoding activities they would be doing during the study [S1 Text]. The memory test portion of the study was omitted from this explanation, only being mentioned as a "third activity". Before beginning the encoding task, instructions were reviewed with the participant.

Encoding stimuli consisted of 4 blocks of 40 color images of commonly known, visually distinct, objects overlaid on top of one of two backgrounds (Fig 1). Object images fell under 1 of 7 categories (animal, clothing, fruit, vegetable, toy, tool, instrument), and were approximately



**Fig 1. Encoding task example.** Example images shown to the participants taken from the encoding task. The leaf symbols indicate the "living/non-living" question (semantic encoding). The face symbols indicated the "I like it / do not like or do not care about" question (self-referential encoding).

320x320 pixels. Background images consisted of a forest background and a beach background, both 800x640 pixels with similar pixel intensity range. Below each background were two symbols indicting the question the participant must answer. The two potential questions were "Do you like this object or dislike/not care about it?", indicated by a smiling cartoon face and a neutral cartoon face, or "Is this a living or not living object", indicated by a leaf and a leaf with a red X through it. The positive options (smile or leaf) were always displayed on the left half of the screen, while the negative options were always displayed on the right. This "like/do not like" paradigm was similar to methods used by several previous studies on self-referential effects. [9, 26, 27]. The background and questions were randomly assigned to each object, ensuring that each category had an equal distribution of the 4 background/question combinations.

The task consisted of 4 blocks of the encoding activity, where the participant would be shown one set of stimuli images inside the scanner. Each image was shown for 3 seconds, followed by a fixation screen, consisting of a while "+" symbol on a black background, for 1 to 9 seconds. Using two buttons, one in each hand, participants were instructed to press the button that corresponded to their answer as quickly and accurately as they could before the next image was shown.

**2.3.2 Kaufman Brief Intelligence Test-2.** Immediately after the memory encoding sessions, participants completed the Kaufman Brief Intelligence Test-2 (Kbit-2). The Kbit-2 was administered by researchers for the purpose of obtaining a brief, reliable, and well-normed assessment of intelligence that measured verbal and nonverbal abilities. A time limit of 30 minutes was placed on the administering of the test in control the amount of time between encoding and testing. If a participant took longer than 30 minutes, the activity was stopped and resumed after the administration of the Memory Testing task. Memory Testing Task.

**2.3.3 Memory testing task.** Upon the completion of the Kbit-2, participants were then tested on what they recalled from the Encoding task. Participants were not informed that they would be tested on their memory of the Encoding task. This task consisted of 3 blocks of 80 images either from the encoding task or new images from the same categories, totalling 240 images (80 new, 160 previously seen during the Encoding task). Images were displayed in pseudorandom order, with no more than three consecutive images from either the new image set or from the encoding activity. For each image, participants were asked to first determine whether they remembered seeing the object during the encoding activity and also remember specific details (e.g., what the image looked like on the screen, what they were thinking at the time etc.), didn't remember seeing the object, or thought the object was familiar but could not confidently recall additional details (denoted by the options "Remember", "New", and "Familiar" respectfully). We have included a copy of the detailed instructions in the [**S1 Text**]. The RK instructions followed closely to Gardiner (1988) [28]. If the participant chose either "Remember" or "Familiar", they were then asked to answer which of the two backgrounds the

object was shown with, followed by being asked which of the two question prompts (living/non-living, or like/don't like) the object was shown with. The test was administered on a laptop with no expressed time limit and broken into 3 blocks. Participants completed a short practice test to make sure they understood the task and researchers monitored the participants for the duration of each block of the test.

## 2.4 Statistical analysis

The trials were categorized based on the answers provided during the testing task into "Hits" (old objects correctly identified as "Remember" or "Familiar"), "Miss" (forgotten objects), "False Alarm" (new objects falsely identified as "Remembered" or "Familiar") and "Correct Rejection" (new objects identified as "New"). Hits were further divided into Remember and Familiar based upon the participants answer. Using the Independence Remember/Know procedure (IRK) [29], familiar rates were estimated by dividing the rate of Familiar responses by 1-(rate of Remember responses). Remember and Familiar trials were further divided into those with correct source information (background image correct, encoding question correct, or both source correct), and those without source information (item-only memory). Source memory were calculated by dividing the number of correct answers by the total number of either correct Remember responses as the denominator. Because source judgements following a Familiar response are likely to contain mostly guesses and there were very few Familiar responses overall ($<$10%), we decided to conduct source memory analyses on Remember trials only. Source accuracy was estimated by subtracting the proportion of Remember trials with incorrect source from trials with both source correct, following Leshikar et al. [30], and Duarte et al. [31]. False Alarm (FA) answers were separated by the erroneous information each participant gave, namely Remember or Familiar, beach or garden background, and self-referential or semantic encoding question. Due to the use of the IRK model for estimation of familiarity, which utilizes the recognition rate in its calculation, the independence of the rates of each memory type is compromised. This questionable independence effects the validity of the corresponding ANOVA, which is why it was not performed on the memory test results.

To further analyse the psychological processes that went into the answers provided by participants, the use of multinomial processing trees (MPT) were implemented. MPT models attempt to estimate latent parameters from observed category frequency counts. In the case of this analysis, the latent parameters represent to the theoretical psychological steps taken by participants when answering a question on the recognition test. The rates at which each participant answered the different questions from the test, grouped by the source information presented with the item, were used as the category frequency counts in the MPT model. The MPT analysis program multitree [32] was used to perform this analysis. Each participant's MPT results were then analysed using t-tests and ANOVA methods to determine significant differences in answering patterns.

## 3 Results

### 3.1 Overall memory performance

Memory accuracy was calculated by subtracting the percentage of False Alarms from the percentage of Hits [33], excluding items in which the participant failed to answer encoding question in time. Across all participants, overall mean accuracy rate of item recognition (Hit-FA) was 0.57 ± 0.16. The mean accuracy rate for Remember trials, calculated by taking the rate of correctly answering Remember and subtracting the rate of False alarms in which the participant erroneously said Remember for a new item, was 0.53 ± 0.17. The mean accuracy rate for Familiar trials (Familiar rate–FA Familiar rate) was 0.16 ± 0.14. Of the New objects displayed

**Table 1. Remember and Familiar trial results for studied items.**

| | Studied Items | | | | | | | | |
|---|---|---|---|---|---|---|---|---|---|
| | Remember | | | | Familiar | | | | Miss |
| | | | | | Unaltered | | Estimated | | |
| **Self** | .70(.20) | | | | .08(.10) | | .24(.22) | | .22(.16) |
| **Living** | .48(.17) | | | | .11(.09) | | .21(.16) | | .42(.16) |
| | *2-source* | *Task-only* | *Background-only* | *0-source* | *2-source* | *Task-only* | *Background-only* | *0-source* | |
| **Self** | .33(.14) | .24(.10) | .08(.05) | .06(.04) | .03(.05) | .02(.03) | .01(.02) | .01(.02) | |
| **Living** | .20(.09) | .15(.07) | .07(.05) | .06(.04) | .05(.05) | .03(.04) | .02(.03) | .02(.02) | |

Raw mean proportion (standard deviation) of Remember and Familiar trials from studied trials. The rate of successful recollection for none, one, or both pieces of source information is displayed for items of Remember and Familiar trials. Estimated familiar rates were calculated using the independent Remember/Know procedure [29].

to the participant during the memory test, the rate of false alarm answers in which the participant said they "Remembered" the object was 0.061 ± 0.073 and rate they said the object was "Familiar" was 0.052 ± 0.06. False alarm rate did not differ between "Remembered" and "Familiar" trials (p = .4).

## 3.2 SRE on recollection versus familiar-based memory

The raw proportion of trials with each memory outcome (Remember, Familiar, Forgotten/Miss for studied items; false alarm and correction rejection for unstudied items) under the self- and semantic- encoding conditions are listed in Tables 1 and 2. Recollection and familiarity memory accuracy scores for item recognition were calculated as the rate of Remember or estimated Familiar responses for the studied items (estimated by the independent RK or IRK procedure) minus the corresponding rate of False Alarms (where an answer of Remember or Familiar was given for the unstudied items).

t-tests of the memory accuracy scores were performed using the IRK estimated familiarity values. Memory accuracy for Remember items was significantly higher in the self-referential encoding condition compared to semantic encoding condition (t(51) = 11.86, p < 0.001) (Fig 2). Memory accuracy for Familiar items was not significantly better when encoded self-referentially compared to those encoded semantically (t(51) = 2.00, p = 0.051). The ratio of the Remember / Familiar rates was significantly higher for self-referential encoding compared to semantic encoding (t(51) = 2.72, p = .009). This larger ratio supports the claim of SRE's effect on memory, indicating the increased detail to which a participant believed that they remembered an item.

**Table 2. Remember and Familiar trail results for unstudied items.**

| | Unstudied Items | | | | |
|---|---|---|---|---|---|
| | Remember | Familiar | | New | |
| | | Unaltered | Estimated | | |
| **Self** | .030 (.04) | .02(.03) | .02 (.04) | | |
| **Living** | .031 (.04) | .03(.04) | .03 (.04) | .89 (.11) | |
| **Total** | .06 (.07) | .05 (.06) | .06 (.07) | | |

Raw mean proportion (standard deviation) of Remember and Familiar trials from unstudied trials. Estimated familiar rates were calculated using the independent Remember/Know procedure [29].

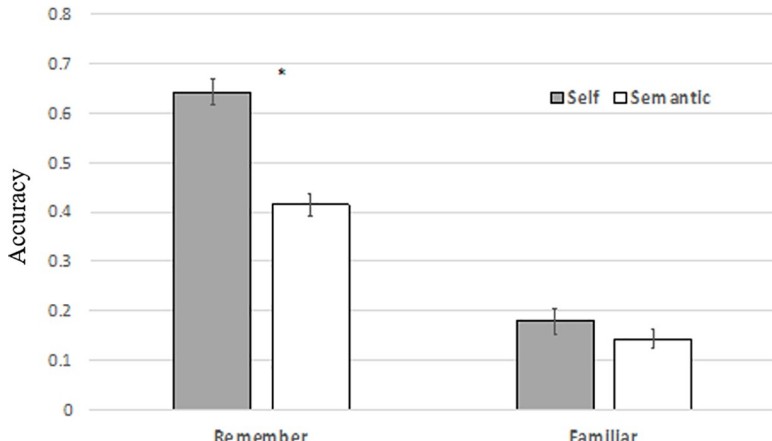

**Fig 2. Remember and Familiar mean accuracy.** Mean memory accuracy rates for "Remember" and "Familiar" trials from the self-referential and semantic encoding conditions. Accuracy was calculated as the rate of "Remember" and "Familiar" trials minus the false alarm rate for "Remember" and "Familiar" respectively. Familiar rate was estimated with the IRK procedure. Error bars denote the standard errors of the mean. * p < .001.

There was no significant difference between the proportion of false alarms that were followed by a self-encoding task judgement compared to false alarms that were followed by a semantic encoding task judgement for either Remember (p = .8) or Familiar (p = .3) false alarms.

## 3.3 SRE on source memory

Source memory analysis was restricted to studied trials with a "Remember" response. Source accuracy for getting both source information (encoding question and background) correct was significantly higher in the self-referential condition compared to the semantic encoding condition (t(51) = 2.44, p = .018) (Fig 3) (Table 3). Self-referential encoding compared to semantic encoding resulted in a significant higher percentage of trials out of the Remember

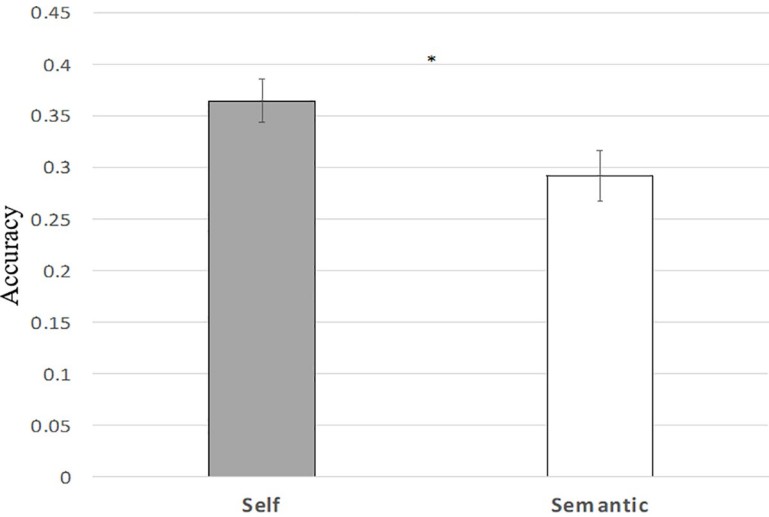

**Fig 3. Mean source accuracy.** Source accuracy estimate for getting both source information correct for Remember responses from studied trials, categorized by encoding condition (Self vs Semantic). Error bars denote the standard errors of the mean. * p < 0.05.

**Table 3. Mean source memory.**

|  | 2-source | Encoding Question | Background | Item-only |
|---|---|---|---|---|
| **Self** | .45(.11) | .77(.12) | .58(.09) | .08(.37) |
| **Semantic** | .42(.12) | .71(.14) | .56(.09) | .13(.59) |

Mean proportion (standard deviation) of trials with correct source and incorrect source (item-only) in studied trials recognized as "Remember".

trials with correct source (one or both source correct) or lower percentage of item-only trials (incorrect source) (t(51) = 2.81, p = 0.007).

Across both the self- and semantic encoding conditions, the proportion of trials with correct judgement on the background image was lower than trials with correct encoding question (t(51) = 9.26, p < .001). The proportion of trials with correct response for encoding question was significantly greater in the self-encoding condition compared to the semantic encoding condition (t(51) = 2.46, p = 0.0175) (Fig 4). The percentage of trials with correct judgement of the background image was not significantly greater in the self-encoding condition compared to the semantic condition (t(51) = 0.728, p = 0.470) (Fig 4).

### 3.4 False alarms

A two-factor repeated measures ANOVA of the answers provided by participants during a False Alarm showed that there was no significant interaction between memory type (Remember vs. Familiar), background source (beach vs. garden), or encoding condition (Self vs. Semantic) (F (51) = 0.627, p = 0.432). False Alarm rates were calculated by dividing the number of a given answer combination (Remember/estimated Familiar, Beach/Garden, Self/Semantic) divided by the total number of FA of the given memory type. None of the factors had a significant main effect, and a t-Test of the effect of the participant's answers for encoding question found no significant difference (Table 4). These results support the idea that there is no significant bias in the participants' answering patterns, and thus reinforce the accuracy of the data collected.

### 3.5 Reaction time

During encoding, reaction time for self-referential trials was slower than the semantic condition (t(51) = 9.57, p < .001). Mean reaction time was 1441ms ± 235.6 for the self-referential

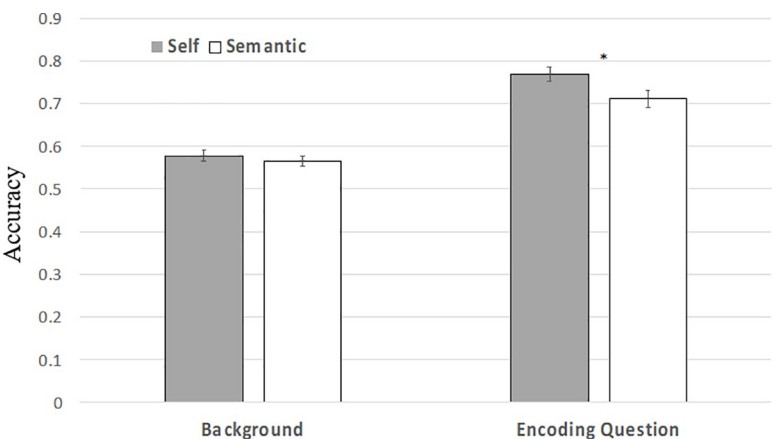

**Fig 4. Mean background accuracy.** Mean proportion of trials with correct judgement of the background image or encoding question in Remember trials. Error bars denote the standard errors of the mean. * p < 0.05.

**Table 4. False alarm results.**

|  | Self | Semantic | p (Self vs. Semantic |
|---|---|---|---|
| **Remember-Garden Background** | 15.97 ± 3.40 | 13.98 ± 3.23 | 0.598 |
| **Remember Beach Background** | 12.51 ± 2.26 | 14.20 ± 3.73 | 0.623 |
| **Familiar_Garden Background** | 8.82 ± 2.47 | 12.23 ± 2.47 | 0.318 |
| **Familiar_Beach Background** | 9.99 ± 2.53 | 11.48 ± 4.16 | 0.691 |

t-Test results for the effect of encoding question on False Alarm (FA) answer rates as percentages. The rate of answering a specific combination was calculated by dividing the quantity by the total number of FA. No significant difference was observed between a participant's likelihood to select the self or semantic answer based on their answer for memory type and background.

condition and 1283ms ± 229.2 for the semantic encoding condition. There was no significant correlation between reaction time and memory accuracies (ps > .2).

## 3.6 MPT analysis

Two MPT models were created for each participant, each modelling the same test results but with the location of both types of source memory in the tree swapped [S2 Text]. A t-test of the MPT results for each participant found no significant trend in answering patterns of participants for erroneously recollected items (False Alarms). The rate at which each participant answered a given combination of source memory answers was not statistically more likely than any other combination, denoting the lack of a bias. ANOVA analysis of the False Alarm results also found no significant interaction between the answering patterns.

Significant differences were noted between the initial recognition rate of an item, i.e. whether they answered that a previously encoded item was New, based on the encoding prompt. These differences were observed between the recognition rates of items self-referentially encoded with a garden background (0.781 ± 0.166) and items semantically encoded with both the garden (0.588 ± 0.187) and beach (0.578 ± 0.151) background (t(51) = 9.771, p < 0.001 and t(51) = 9.956, p<0.001, respectfully). The same trend was seen between items self-referentially encoded with the beach background (0.783 ± 0.172) and items semantically encoded with both the garden and beach background (t(51) = 9.555, p < 0.001 and t(51) = 10.535, p<0.001, respectfully). There was notably no significant difference between the recognition rates of items with the same encoding prompt, regardless of background. This was reflected in the ANOVA test of recognition which found that the encoding prompt had a main effect on the rate of recognition (F(51) = 132.136, p < 0.001), while the background did not.

Using the two unique MPT models, we were able to observe any significant interactions between the two types of source information given the accuracy of the participant in correctly answering the other source. The results of said conditional probabilities can be found in Table 5 and Fig 5. ANOVA analysis of the different conditional probabilities found no significant interaction between the background and prompt presented during encoding on the success rate of a source given the other source success.

A t-test between the rate of successfully recollecting the background given the encoding prompt was successfully or unsuccessfully recollected found almost no significant difference in outcome based on the background and prompt displayed during encoding. The exemption of this trend were items semantically encoded with the beach background. The rate of correct background recognition was significantly higher when the prompt was also successfully recollected (t(51) = 2.335, p = 0.024). Items encoded with this background and prompt also had significantly higher rate of correctly recollecting the prompt given the background was successfully recollected (t(52) = 2.361, p = 0.022). No other significant difference was observed

**Table 5. MultiTree results.**

| | Source Combination | | | |
|---|---|---|---|---|
| **Remember** | **Garden & Self** | **Garden & Semantic** | **Beach & Self** | **Beach & Semantic** |
| $B_{correct}$ **if** $P_{correct}$ | 0.61 ± 0.17 | 0.62 ± 0.20 | 0.56 ± 0.17 | 0.54 ± 0.21 |
| $B_{correct}$ **if** $P_{incorrect}$ | 0.68 ± 0.23 | 0.60 ± 0.29 | 0.55 ± 0.31 | 0.43 ± 0.28 |
| $P_{correct}$ **if** $B_{correct}$ | 0.73 ± 0.17 | 0.70 ± 0.23 | 0.84 ± 0.13 | 0.78 ± 0.19 |
| $P_{correct}$ **if** $B_{incorrect}$ | 0.77 ± 0.22 | 0.70 ± 0.25 | 0.80 ± 0.18 | 0.70 ± 0.22 |
| **Familiar** | **Garden & Self** | **Garden & Semantic** | **Beach & Self** | **Beach & Semantic** |
| $B_{correct}$ **if** $P_{correct}$ | 0.52 ± 0.27 | 0.56 ± 0.32 | 0.50 ± 0.30 | 0.49 ± 0.32 |
| $B_{correct}$ **if** $P_{incorrect}$ | 0.50 ± 0.23 | 0.52 ± 0.31 | 0.53 ± 0.30 | 0.49 ± 0.32 |
| $P_{correct}$ **if** $B_{correct}$ | 0.62 ± 0.26 | 0.65 ± 0.31 | 0.52 ± 0.25 | 0.59 ± 0.35 |
| $P_{correct}$ **if** $B_{incorrect}$ | 0.62 ± 0.29 | 0.60 ± 0.36 | 0.56 ± 0.24 | 0.55 ± 0.34 |

Results for the effect of the rate of successful recollection of one source type given the other source information is either correct of incorrect. The "B if P" or "P if B" denote the rate at which a source is correct given the other source is correct/incorrect, with B representing background and P representing encoding prompt. For a visual representation of the multitree models, see Fig 5.

in the success rate of correct prompt recollection given the background was also correctly recollected for items displayed with the other combinations of backgrounds and prompts.

Due to the large quantity of results from the MultiTree analysis, the.mpt files used in the calculation, containing the results of each participant's model, can be found in the OSF repository in the MPT folder. In that folder is also a detailed explanation of how to read the.mpt file notation.

# 4 Discussion

We investigated the interaction of the subjective recollection experience and the self-referential effect on source memory. Self-referential benefits on memory accuracy were observed in recollection but not in familiarity-based memory. Self-referential facilitation on familiarity was marginal and only trended toward statistical significance. This SRE facilitation on memory accuracy extended to source memory accuracy. In regards to the two different types of source information, self-referential encoding resulted in better recollection of the encoding context, but did not facilitate the recollection of a peripheral source (background image) which was not tied to the encoding task. This difference in source memory between the two encoding conditions supports the idea that SRE facilitates recollection of source information that is explicitly processed during the encoding episode. As this benefit is not extended to peripheral source memory, different mechanisms may be utilized when encoding this information.

The encoding method also significantly affected the ratio of Familiar to Remember judgements, shown through the difference between the rate of each memory type, even after the use of the IRK method to estimate familiarity, similar to what was seen in Conway and Dewhurst [24]. Self-referentially encoded objects had a higher chance of being judged as "Remember" than objects semantically-encoded, suggesting self-referential encoding enhances subjective recollection. Our findings replicate those of previous studies, with the overall item recognition (regardless of source correctness) for self-referentially encoded stimuli significantly higher than that of semantically encoded stimuli. Consistent with prior investigations [21, 30], this SRE facilitation for memory was observed in recollection but not in familiarity-based memory.

Our analysis of false alarms suggests that there were no biases in selecting either encoding question or either of the background images. It was not more likely to attribute the encoding context to the self-referential condition after a Remember judgement was made. This was

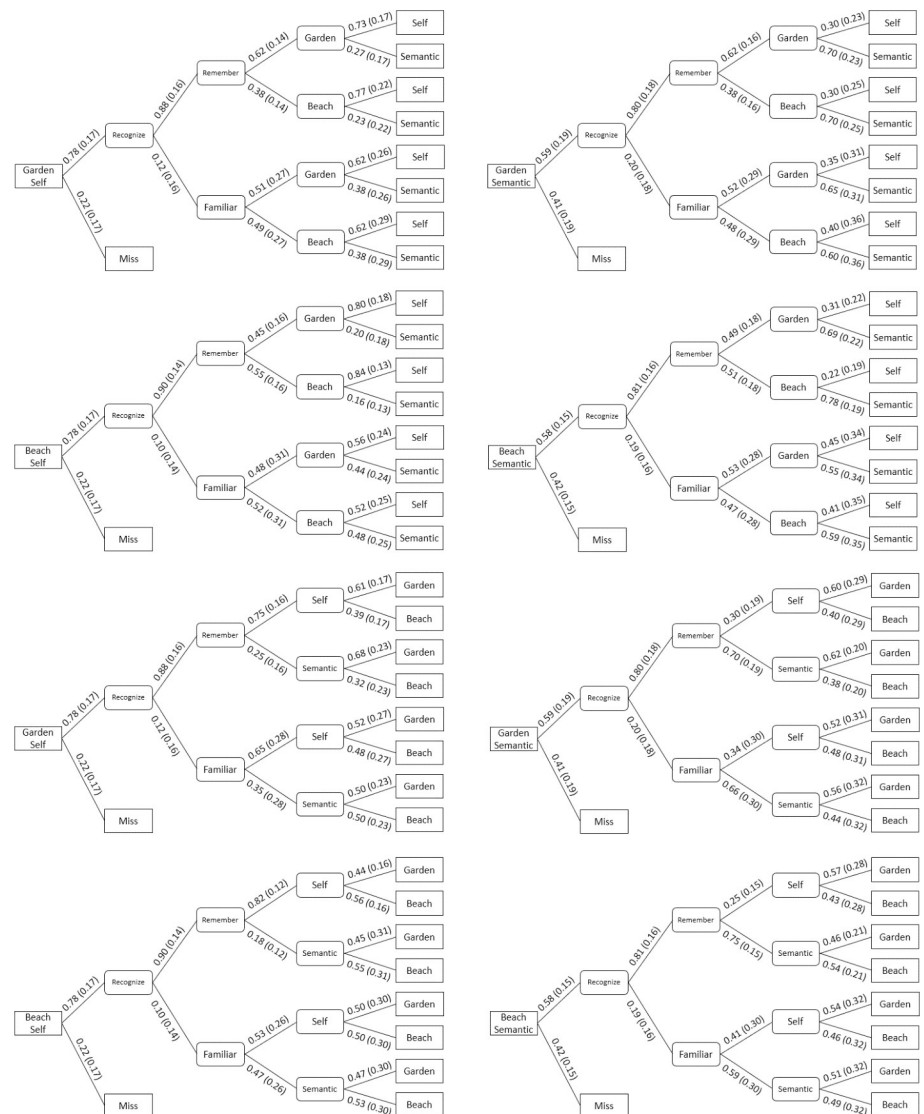

**Fig 5. MPT conditional probabilities.** Mean (standard deviation) conditional probabilities of all participants' test results, organized by source memory. Each box represents an answer option on the memory test.

supported by the results of our Multi-tree analysis, which found no significant trend in the answering patterns of participants for false alarms.

Our MPT analysis found that the success rates for each source type were largely independent from each other, with no significant influence on success rates present between sources. The likelihood of a participant correctly recollecting one piece of source information did not significantly vary based on whether they had correctly or incorrectly recalled the other source information. This alludes to a separation of peripheral source memory and memory of required source information. Not all types of source memory benefit from the self-referential effect. The encoding method did, however, have a significant effect on the rate of item recollection. Items encoded self-referentially had a higher rate of recollection, independent of the background displayed with the item.

Our findings contribute to and expand upon the knowledge of SRE in several ways. First, our findings suggest that self-referential encoding has different effects on source memory in

recollection and familiarity-based memory. The majority of previous SRE studies on source memory have not differentiated recollection vs familiarity-based memory. It is possible, without separating out familiarity-based memory, the SRE on source memory can be attenuated, which could have contributed to some of the inconsistencies in previous studies. Second, we used emotionally neutral images in our study. Contrary to the work of Durbin, et al. [14], which only found SRE in positive pictures, we observed a beneficial SRE in source memory for neutral stimuli. A potential cause of this difference in findings could come from the nature of the task, with Durbin et al. asking if an adjective was self-describing, while we asked participants whether they like/don't like an object. Third, our study, to the best of our knowledge, is the only to uniquely include source information that is not directly tied to the encoding activity. Participants were not asked any question about the background image or asked to pay attention to it, resulting in a source memory metric not directly intertwined with item recollection by necessity to complete the encoding activity. Our results suggest that the self-referential benefits on source accuracy was only restricted to the source information directly tied to the task or information that participants were explicitly processing during the task.

Reaction time for the self-referential encoding was significantly longer than semantic encoding. However, reaction time did not correlate with any metric of memory accuracy, unlike the significant correlation between encoding prompt and reaction time that has been observed in other studies [17]. This suggests that SRE could not be simply explained by the length of exposure to the stimuli. Instead, our results and other previous findings suggest that self-referential encoding creates a richer, more detailed memory trace compared to semantic encoding that can be recollected on later.

There are a few limitations to this study that should be noted. First, our study did not manipulate the valence of the stimuli. Although most of our stimuli were neutral, a small percentage of them could be perceived as positive in valence for certain individuals (e.g., a sunflower). Therefore, we could not completely rule out the possibility that some positively-valanced stimuli were contributing to the SRE effect. However, we believe that this influence is most likely not significant due to the small portion of the stimuli that could be perceived as positive in valence. Second, it was possible that some unusual association between the object and the background (e.g., a fish being shown on a forest background) helped the memory for those trials due to their novelty. However, we do not believe that the SRE we found were influenced by this as the object-background pairing was random and the two background images were evenly distributed across the two encoding conditions. Post-hoc inspection of the stimuli identified about 20 out of the 160 images with potentially "odd" pairing, with 9 in self-referential condition and 10 in semantic condition.

In summary, our investigation into the interaction between SRE and recollection/familiarity-based memory found both the predicted universal improvement in item memory due to SRE and unique results with source memory. Our findings suggest that self-referential encoding facilitates recollection of source information that is explicitly processed during the encoding episode. This facilitation is not extended to peripheral information, and may denote a separation in memory mechanisms. These results suggest self-referential encoding creates a richer, more detailed memory trace which can be recollected later on, which also improves one's own judgment of their memory capabilities.

## Supporting information

**S1 Text. Procedure and script for participants.** Provides the steps and script used while conducting the experiment.
(DOCX)

**S2 Text. Guide for interpreting MultiTree model.** Explains the labeling structure used for the MultiTree models used in the MPT analysis.
(DOCX)

## Acknowledgments

We thank Pat Ourand, Jessica Cheng and Melissa Eustache for their help in subject recruiting and testing, and Cristiana Camardella for her support in subject recruiting. We'd also like to thank Terri Brawner, Kathleen Kahl, and Ivana Kusevic for assistance with running participants.

## Author Contributions

**Conceptualization:** Xiaoqian J. Chai.

**Data curation:** Ross Lawrence.

**Formal analysis:** Ross Lawrence.

**Funding acquisition:** Xiaoqian J. Chai.

**Investigation:** Xiaoqian J. Chai.

**Methodology:** Xiaoqian J. Chai.

**Software:** Ross Lawrence.

**Supervision:** Xiaoqian J. Chai.

**Writing – original draft:** Ross Lawrence.

**Writing – review & editing:** Xiaoqian J. Chai.

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
