## [Decision Letter · Decision Letter 0]

15 Apr 2020

PONE-D-20-04302

Self-referential encoding of source information in recollection memory

PLOS ONE

Dear  Dr. Lawrence,

Thank you for submitting your manuscript to PLOS ONE. After careful consideration, we feel that it has merit but does not fully meet PLOS ONE’s publication criteria as it currently stands. Therefore, we invite you to submit a revised version of the manuscript that addresses the points raised during the review process.

Both reviewers and myself agree that the topic of your paper is very interesting. Your introduction highlights the novelty of the work. The major problem concerns the analyses and presentation of results.  You have analysed remember and know responses within the same ANOVA.  This approach is problematic as these two measures are not independent.  Both reviewers have suggested an alternate approach for analysing this data based on methods used by Yonelinas & Jacoby, 1995. Reviewer 2 raises questions about how source memory was calculated. Reviewer 2 has made further suggestions for analyses which could strengthen your paper. based on Boywitt et al. (2012; Psychology and Aging.  Further reviewer one asks you to justify why guess responses were not analysed. There are problems with how you  presented your data in Table 1 as well as your figures.  In this type of research it is also extremely important that your methods are well-described.  Both reviewers and myself felt that the description fo your methods could be more detailed.  In your revision you must address the analysis issues raised by both reviewers as well as the other issues raised.   

We would appreciate receiving your revised manuscript by June 30 2020. I hope that you, your colleagues and family are doing well.  if you need more time, particularly in light fo the Covid- 19 crisis, please  let us know.. To enhance the reproducibility of your results, we recommend that if applicable you deposit your laboratory protocols in protocols.io, where a protocol can be assigned its own identifier (DOI) such that it can be cited independently in the future. For instructions see: http://journals.plos.org/plosone/s/submission-guidelines#loc-laboratory-protocols

We look forward to receiving your revised manuscript.

Kind regards,

Barbara Dritschel, PhD

Academic Editor

PLOS ONE

Journal Requirements:

2. Thank you for including your ethics statement: "Johns Hopkins School of Medicine IRB Approval Number: IRB00151734. Form of consent: Written"

Please amend your current ethics statement to confirm that your named institutional review board or ethics committee specifically approved this study.

"This study was supported by the Johns Hopkins Cognitive Neurology Departmental Fund to XJC."

"X.C.:

Therapeutic Cognitive Neuroscience Fund

Grant Number: 80026224

URL: N/A

Reviewers' comments:

Reviewer's Responses to Questions

**Comments to the Author**

1. Is the manuscript technically sound, and do the data support the conclusions?

Reviewer #1: No

Reviewer #2: Partly

2. Has the statistical analysis been performed appropriately and rigorously? 

Reviewer #1: No

Reviewer #2: No

3. Have the authors made all data underlying the findings in their manuscript fully available?

Reviewer #1: No

Reviewer #2: Yes

4. Is the manuscript presented in an intelligible fashion and written in standard English?

Reviewer #1: Yes

Reviewer #2: Yes

5. Review Comments to the Author

Reviewer #1: There are both strengths and weaknesses in this paper. It is an interesting research question that may help explain some autobiographical memory (assuming that much of our autobiographical memory is based on self-referential processing). However, I don’t think the paper should be published in its current form. I note the reasons for my opinion below, in the spirit of improving the research.

The most important issue is that R and K (or F) responses cannot be entered into a single ANOVA with response type as a factor. R and K responses are mutually exclusive, so they are not independent from each other and therefore violate the assumptions of ANOVA. The only way they can be compared to each other is by using the IRK (independent RK) method described by Yonelinas & Jacoby, 1995 (The relations between remembering and knowing as bases for recognition: Effects of size congruency). This creates a probability estimate of familiarity. As a result of the way the analyses were done, it is difficult to review the conclusions because it’s possible that the results will change. Regardless of the publication decision here, these data have to be reanalyzed before submitting again.

Pleasantness judgments are typically thought of as deep processing, rather than being self-referential. Is there evidence about this? That is, I was under the impression that self-referential processing produced better memory than pleasantness judgments.

Many people who use the RK procedure actually use and advocate for the RKG (guess) method. Was there a specific reason to exclude the guess response?

The RK instructions need to be described in a lot of detail, given that brief instructions without practice can lead to a confusion of how to use the responses. A copy of the instructions would be appropriate (perhaps in an appendix if they are long). This will be important for further review of the manuscript.

Having a source question after an F judgment implies to subjects that they should/could have source available. By definition of the procedure though, remember responses include details and know responses do not. If there is recollected detail in a know response, it’s arguable that the procedure is just not being followed correctly. This has implications for whether or not we can conclude that subjects had some accurate source memory (and that it differed between conditions) for familiarity-based memory. This should be discussed.

p should be reported as p = .xxx, unless it’s less than .001, which can be reported as p < .001.

Reviewer #2: Review of “Self-referential encoding of source information in recollection memory”

First and foremost, I wish the authors well in this unprecedented time and hope that they and their families remain safe and healthy.

This paper presents a single experiment examining the effect of self-referential encoding on item and source memory. Participants incidentally encoded objects overlaid on a scene background and were prompted to make a self-referential (pleasantness) or semantic (living/nonliving) judgment for each trial. At test, participants first made a Remember/Familiar/New judgment for old and new objects. Objects attracting a R or F responses were further probed for source memory of the peripheral detail (background scene) and central detail (self vs. semantic). Self-referential encoding increased the accuracy of Remember responses and decreased the accuracy of Familiar responses. Self-referential effects were observed for the peripheral and central details, which was particularly evident for Remember trials. These findings are taken as evidence that self-referential encoding creates a more elaborate and rich memory trace that supports recollection for many different features of an event.

The Introduction does a good job outlining the gaps in the literature about the effects of self-referential encoding on source memory. The experiment was also well designed and to my read there are no major flaws in the design that would confound interpretation of the results. I particularly like the inclusion of peripheral versus central source details in the design, which does help expand research on this topic. I do have some concerns regarding the analysis, particularly the calculation of memory measures, and interpretation of the results that I express below, along with some minor suggestions.

Major Points:

1) The proportion of Remember and Familiar responses are directly compared in the item memory analysis, which is problematic because these proportions are not independent of one another. A higher proportion of R responses reduces the highest proportion of F responses that can be observed. With this analysis approach, it is difficult to determine if different encoding tasks are leading to the trade off or if the difference is simply a statistical artifact. To be clear, this criticism does not apply to the self-referential encoding advantage on the R Hit-FA score reported in the paper, but mainly to the F accuracy measure. I would recommend recomputing the familiarity measures using the Yonelinas & Jacoby (1995; Journal of Memory and Language) independent remember/know correction to provide estimates of recollection and familiarity. I computed these measures using the mean data reported in the paper, and it seems as though self-referential encoding increases both recollection and familiarity relative to semantic encoding. The corrected familiarity measure using the Yonelinas & Jacoby formulas was .207 and .148 for self-referential and semantic encoding, respectively. However, whether this holds in the individual data remains to be seen. I do believe addressing this is something that will add to the literature and increase the potential impact of this paper.

2) I was unclear as to how source memory was calculated. It seems to be calculated based on the number of trials with the correct source judgment divided by the total number of old trials (separately the self-referential and semantic types). That is, for example, for the ‘Familiar’ source accuracy in the self-referential encoding task, source memory for the background scene was determined as the number of trials receiving an accurate ‘Familiar’ and source memory response divided by the number of old items in self-referential encoding. This is the only way the numbers shown in Table 1 and the Figures make sense, given they are quite low (which would be below chance). Source memory should not be calculated in this way because it confounds item memory with source memory. The most appropriate way to compute these source accuracy measures is to use the number of correctly identified items (e.g., all ‘familiar’ hits) as the denominator and the numerator as the number of correctly identified items labeled familiar that also attracted an old response. Otherwise, it cannot be easily discerned if the source memory results are just a reflection of the pattern of results observed in the item memory measure.

Minor Points:

3) The data reported in Table 1 is not entirely clear. The Remembered (R) data is not the raw proportion of remember responses but is instead the Hits – FA for R responses. This should be made clear, as well as the other columns. I would also recommend including a table of the total proportions of each cell formed by crossing RKN response with source accuracy (e.g., both correct, encoding task only, background scene only). This will help readers be able to make additional comparisons in the group data and will be helpful in identifying potential response bias effects (i.e., were false alarms more likely to be followed by a self-referential task judgment relative to a semantic task judgment). In fact, it would be helpful to report source misattributions.

4) The y-axis on the figures can use a line with tick marks. It was difficult to determine the values. Also, the y-axis label ‘% correct’ is a little misleading. Hits – FA for R and F are not % correct, but instead are accuracy or discriminability measures.

5) The presentation of the source memory results was confusing. The post-hoc comparisons for the background and encoding task measures were reported in the ‘Both Source Correct’ section. These should be moved to their respective sections to facilitate ease of reading.

6) What version of the RF instructions were used? There is a lot of variability in this task, and including the instructions in the OSF repository (or as an appendix in the paper) will help readers understand how the task was done and allow for better replication of the method.

Suggestion:

7) My last comment is not something that needs to be addressed unless the authors wish to pursue it. It is simply a suggestion based on a thought I had while reading the paper. As I noted above, a strength of this paradigm is the inclusion of multiple source details. As the authors noted, self-referential encoding may lead to the formation of richer memory traces that support subsequent recollection, as evident by the reported increase in the proportion of trials where both sources was remembered. One way to formally test this would be to use multinomial processing tree model of the RKN and source memory data similar to Boywitt et al. (2012; Psychology and Aging). This approach allows for estimation of a parameter that reflects the degree to which multiple source detail are recollected bound or independent of one another (separately for R and F trials). I think this would be interesting and would help get around the issue I mentioned above regarding estimation of source accuracy measures and potentially provide an informative insight into how self-referential encoding supports recollection (i.e., by increasing binding amongst multiple elements, both central and peripheral, to an event).

6. PLOS authors have the option to publish the peer review history of their article (what does this mean?). If published, this will include your full peer review and any attached files.

Reviewer #1: No

Reviewer #2: No

---

## [Author Response · Author response to Decision Letter 0]

5 Nov 2020

Full response to each reviewer can be found in "Response to Reviewers.docx"

---

## [Decision Letter · Decision Letter 1]

8 Dec 2020

PONE-D-20-04302R1

Self-referential encoding of source information in recollection memory

PLOS ONE

Dear Dr. Lawrence, 

Thank you for submitting your manuscript to PLOS ONE. After careful consideration, we feel that it has merit but does not fully meet PLOS ONE’s publication criteria as it currently stands. Therefore, we invite you to submit a revised version of the manuscript that addresses the points raised during the review process

You have done a good job in addressing points raised by both reviewers.  However there are some additional changes that are required before your manuscript can be accepted.  You need to make adjustments to Tables 1 and 4 .  For example Table 1 contains a lot of information and so two tables may be a better option.  The  description of the results could be clearer. You need to state whether all analyses are IRK based analyses. The discussion could be expanded to discuss if differences between source  memory as a function of conditions has implications for our understanding of source memory in general. Reviewer1 raised  some further points which need to be addressed.

Please submit your revised manuscript by  January 7 2021. If you will need more time than this to complete your revisions, please reply to this message or contact the journal office at plosone@plos.org. Please include the following items when submitting your revised manuscript:

We look forward to receiving your revised manuscript.

Kind regards,

Barbara Dritschel, PhD

Academic Editor

PLOS ONE

Reviewers' comments:

Reviewer's Responses to Questions

**Comments to the Author**

1. If the authors have adequately addressed your comments raised in a previous round of review and you feel that this manuscript is now acceptable for publication, you may indicate that here to bypass the “Comments to the Author” section, enter your conflict of interest statement in the “Confidential to Editor” section, and submit your "Accept" recommendation.

Reviewer #1: (No Response)

Reviewer #2: All comments have been addressed

2. Is the manuscript technically sound, and do the data support the conclusions?

Reviewer #1: Partly

Reviewer #2: Yes

3. Has the statistical analysis been performed appropriately and rigorously? 

Reviewer #1: Yes

Reviewer #2: Yes

4. Have the authors made all data underlying the findings in their manuscript fully available?

Reviewer #1: Yes

Reviewer #2: Yes

5. Is the manuscript presented in an intelligible fashion and written in standard English?

Reviewer #1: Yes

Reviewer #2: Yes

6. Review Comments to the Author

Reviewer #1: The authors have dealt with my concerns sufficiently. I have just a few more points.

The supplementary material is sufficient (in fact, you could probably just include the study and test instructions). I think that one limitation that should be mentioned is that there was no practice with feedback session with the R and F responses prior to beginning the test.

Table 1 is hard to read. Try reorganizing it (maybe break into two tables?).

Based on the language used, it’s hard to tell if all the analyses are restricted to IRK-based analyses or if there are still some F-based analyses in there. When referring to the IRK-based values, refer to them as familiarity estimates (also, corrected remember rates should also be called remember estimates). I found this on page 12 and again on page 15 for the false alarm section.

Table 4 is well-organized, but the information provided (Pcorrect if Bcorrect) is hard to picture. An actual figure of the MPT with the values would be far better than the table. Even if the program doesn’t output a figure, it can easily be done even in powerpoint (although it is a bit time-consuming).

The idea that the ratio of Remember to Familiar judgment changed across conditions seems repetitive in that the ANOVA showed that there were differences, and so this second analysis seems unnecessary. This comes through especially in the second paragraph in the discussion. If this is adding information above and beyond the ANOVA with R and F estimates, then spell it out explicitly so the reader can’t be confused.

What does the difference in source memory between conditions tell us about source memory in general?

Reviewer #2: Review of “Self-referential encoding of source information in recollection memory”

I want to thank the authors for their careful consideration of my prior comments. They addressed all of them and I believe the manuscript is now much clearer and stronger. These results are indeed interesting and expand on our understanding of the SRE effect.

I only have one relatively minor comment related to the analysis of IRK estimates of recollection and familiarity. I still do not think it appropriate to include both measures in an ANOVA (though I am sure other papers have done so previously) given that the data are derived from non-independent trials. I would recommend dropping this and just focusing on the t-tests presented to decompose the interaction. That being said, I do not think anything major hinges on this, and I am happy to let the authors determine whether or not they want to address it.

7. PLOS authors have the option to publish the peer review history of their article (what does this mean?). If published, this will include your full peer review and any attached files.

Reviewer #1: No

Reviewer #2: No

---

## [Editor Report · Decision Letter 2]

19 Feb 2021

Self-referential encoding of source information in recollection memory

PONE-D-20-04302R2

Dear Dr. Lawrence,

We’re pleased to inform you that your manuscript has been judged scientifically suitable for publication and will be formally accepted for publication once it meets all outstanding technical requirements.

Kind regards,

Barbara Dritschel, PhD

Academic Editor

PLOS ONE
---

## [Editor Report · Acceptance letter]

23 Mar 2021

PONE-D-20-04302R2 

Self-referential encoding of source information in recollection memory 

Dear Dr. Lawrence:

I'm pleased to inform you that your manuscript has been deemed suitable for publication in PLOS ONE. Congratulations! Your manuscript is now with our production department. 

Kind regards, 

on behalf of

Dr. Barbara Dritschel 

Academic Editor

PLOS ONE